# Prospective observational study of prevalence, assessment and treatment of pancreatic exocrine insufficiency in patients with inoperable pancreatic malignancy (PANcreatic cancer Dietary Assessment (PanDA): a study protocol

Lindsay E Carnie ![ORCID],[1] Angela Lamarca,[2,3] Kate Vaughan,[2,3]
Zainul Abedin Kapacee,[2] Lynne McCallum,[4] Alison Backen,[2,3] Jorge Barriuso,[2,3]
Mairéad G McNamara,[2,3] Richard A Hubner,[2,3] Marc Abraham,[1] Juan W Valle[2,3]

Marc Abraham was deceased on 17/06/2019.

For numbered affiliations see end of article.

**Correspondence to**
Professor Juan W Valle;
juan.valle@nhs.net

## ABSTRACT

**Introduction** Pancreatic exocrine insufficiency (PEI) in patients with pancreatic malignancy is well documented in the literature and is known to negatively impact on overall survival and quality of life. A lack of consensus opinion remains on the optimal diagnostic test that can be adapted for use in a clinical setting for this cohort of patients. This study aims to better understand the prevalence of PEI and the most suitable diagnostic techniques in patients with advanced pancreatic malignancy.

**Methods and analysis** This prospective observational study will be carried out in patients with pancreatic malignancy (including adenocarcinoma and neuroendocrine neoplasms). Consecutive patients with inoperable pancreatic malignancy referred for consideration of first-line chemotherapy will be considered for eligibility. The study comprises three cohorts: demographic cohort (primary objective to prospectively investigate the prevalence of PEI in patients with inoperable pancreatic malignancy); sample size 50, diagnostic cohort (primary objective to design and evaluate an optimal diagnostic panel to detect PEI in patients with inoperable pancreatic malignancy); sample size 25 and follow-up cohort (primary objective to prospectively evaluate the proposed PEI diagnostic panel in a cohort of patients with inoperable pancreatic malignancy); sample size 50. The following is a summary of the protocol and methodology.

**Ethics and dissemination** Full ethical approval has been granted by the North West Greater Manchester East Research and Ethics Committee, reference: 17/NW/0597. This manuscript reflects the latest protocol V.8 approved 21 April 2020. Findings will be disseminated by presentation at national/international conferences, publication in peer-review journals and distribution via patient advocate groups.

**Trial registration number** 194255, NCT0361643.

## INTRODUCTION
### The pancreas
The pancreas has two main functions; producing enzymes to digest protein, fat and carbohydrates into smaller molecules that the body can absorb, and producing hormones that regulate metabolism (including the regulation of blood sugar levels (insulin and glucagon) and global regulation of other hormones).[1]

### Pancreatic malignancy: the importance of being fit for treatment
Pancreatic cancer (adenocarcinoma) is known to have a poor prognosis with a very low cure rate; most patients diagnosed will die of the disease. In 2014, around 41 000 pancreatic cancer-related deaths occurred in Europe.[2]

### Strengths and limitations of this study

► This prospective study is a first-of-its-kind aiming to better define pancreatic exocrine insufficiency (PEI), its diagnosis and treatment, in patients with inoperable pancreatic malignancy.

► Findings from the demographic cohort will define the prevalence of PEI in patients with inoperable pancreatic malignancy, while the diagnostic cohort will define the most suitable test/panel to define PEI in this setting.

► Results will be validated in the follow-up cohort, including impact on patients' quality of life.

► Due to the nature of PEI and the fact that PEI treatment is considered standard of care, this is a non-randomised study in which all patients will be exposed to PEI treatment (if required) and dietitian input.

► We expect limited statistical power and capacity to assess impact of PEI-related intervention on quality of life and patient outcome derived from characteristics of the study population, study design and limited sample size.

BMJ

The physical location of the tumour can prevent the digestive regulatory functions of the pancreas, causing the systemic symptoms that the majority of patients present with. Symptoms include anorexia (83%), asthenia (86%) and weight loss (85%).[3] Symptoms can impact on quality of life (QoL), nutritional status and performance status (PS), which subsequently may preclude active treatment options such as chemotherapy.[4]

Only approximately 20% of patients are suitable for surgery at diagnosis; these patients undergo pancreatic resection followed by adjuvant chemotherapy with fluoropyrimidine-based or gemcitabine-based treatment.[5–7] A good nutritional status, prior to adjuvant chemotherapy increases the likelihood of a patient completing chemotherapy, which in turn impacts on survival.[8]

Most patients (80%) present with advanced disease and are unsuitable for surgery. Instead, they will receive palliative chemotherapy, aiming to improve QoL and prolong overall survival (OS). Single-agent gemcitabine has long been considered standard of care in patients with a poorer PS, providing a median OS of 6 months.[9] Recent chemotherapy combinations show improved results, reaching a median OS of 8.5 months (nab-paclitaxel/gemcitabine),[10] and 11.1 months (a 5-fluorouracil, oxaliplatin and irinotecan combination).[11]

A retrospective analysis of patients with advanced pancreatic cancer referred to The Christie NHS Foundation Trust found around 40% were not fit for active treatment due to poor baseline PS as per The Eastern Cooperative Oncology Group-PS (ECOG-PS) definition.[12]

The scenario for patients diagnosed with pancreatic neuroendocrine tumours (PanNETs) differs significantly. Prognosis is measured in terms of years, with an estimated median OS of 3.6 years[13] and multiple options of systemic therapy are currently available.[14] The prevalence of PanNETs is rare, with an estimated incidence of 0.8 per 100 000.[13] Whilst the prognosis of these patients is better, this longer survival time means that identifying and minimising the impact of nutritional deficiencies and issues is of particular importance.[15]

## Pancreatic exocrine insufficiency: causing malnutrition in patients with pancreatic malignancy

Pancreatic exocrine insufficiency (PEI) is defined as 'a reduction in pancreatic enzyme activity in the intestinal lumen to a level below the threshold required to maintain normal digestion'.[16]

A high prevalence of PEI has been described in patients with resected (>80%)[17] or advanced disease (92%)[18] in prospective series, and this negatively impacts on QoL.[19] Different mechanisms have been postulated for the development of PEI, including loss of functioning pancreatic parenchyma (by tumour infiltration or resection or concurrent/prior pancreatitis) and/or pancreatic duct obstruction. PEI, leading to maldigestion, steatorrhoea and malnutrition, has been proposed as a leading cause

for the high number of patients with pancreatic malignancy being unfit for active treatment.[20]

Whilst healthcare professionals seem aware of the importance of diagnosing and treating PEI in patients after pancreatic resection, it is often overlooked in patients with advanced disease. This under-recognition and undertreatment of PEI in patients with advanced disease is an ongoing issue, requiring urgent action.[21]

Weight loss is a poor prognostic factor in patients with both resectable and advanced pancreatic malignancy.[22 23] However, little published information exists on the extent of nutritionally mediated weight loss, how this relates to the cancer and how much could be mitigated with proactive pancreatic enzyme replacement therapy (PERT).

### Diagnosis of PEI in patients with pancreatic malignancy

Waiting for symptom development, including steatorrhoea (defined as excess fat in faeces that appears when 90% of pancreatic function is lost) delays the diagnosis of PEI and negatively impacts on nutrition and QoL.[24] Early assessment of exocrine function is fundamental, and should be considered in all patients diagnosed with pancreatic disorders, including cancer.[25]

Diagnosing PEI in patients with pancreatic malignancy can be difficult, and a lack of consensus remains for the optimal assessment method. While 3-day faecal fat quantification is 'gold-standard' for diagnosing PEI, its use in clinical practice is challenging.[16] The secretin test is invasive and has potential for clinical complications, reducing its appeal.[26] Measurable reduction of pancreatic parenchymal thickness in imaging correlates with changes assessed using a $^{13}$C-mixed triglyceride breath test ($^{13}$C-MTBT), with good sensitivity and specificity after pancreatic resection.[27] This has become the new 'standard', replacing the 3-day faecal fat test. The use of current diagnostic techniques such as faecal elastase-1 (FE-1),[28] (postulated to be more useful in patients who have not undergone resection), the $^{13}$C-MTBT[29] and a nutritional panel of blood-based markers warrant further investigation to clarify their use.[30]

In summary: the optimal diagnostic method for PEI in patients with pancreatic malignancy remains undefined; $^{13}$C-MTBT is considered 'gold-standard' but is challenging to apply in daily clinical settings. This study aims to design the most appropriate and least-invasive diagnostic panel, with $^{13}$C-MTBT as the comparator for patients diagnosed with pancreatic malignancy (including both adenocarcinoma and neuroendocrine tumours).

### Treatment of PEI and its impact on quality of life and survival

Guidelines for the management of PEI exist,[31–33] and two publications support using high-dose PERT to mimic the physiological situation, to normalise nutritional status.[29 34] Using a proton pump inhibitor to increase gastric pH, enhancing the efficacy of PERT (by reducing gastric acid-induced enzymatic degradation) in selected patients has also been demonstrated.[35]

At The Christie NHS Foundation Trust, 183 patients with pancreatic malignancy were retrospectively analysed and it was demonstrated that patients receiving nutritional intervention (PERT, nutritional supplements or dietitian support) seemed to receive more chemotherapy and had a longer OS (10.2 months (95% CI 7.5 to 13.3) vs 6.9 months (95% CI 5.5 to 9.9); HR 0.6 (95% CI 0.4 to 0.9); p=0.015), when adjusted for other variables in the multivariable analysis (type of pancreatic cancer, stage at diagnosis, ECOG-PS and chemotherapy treatment).[12] This study also confirmed that PEI is under-recognised and undertreated in patients with advanced disease. Since this was a retrospective study, it is subject to selection and survival bias. Therefore, whilst results are encouraging, prospective studies are required to evaluate the impact of dietetic intervention (including PERT) on QoL, exposure to anticancer treatment, symptom control and outcomes.

In summary: dietetic intervention, early diagnosis and management of PEI could impact patients' OS. This study aims to prospectively assess the impact of such interventions in patients with pancreatic malignancy.

## AIM

This prospective observational study aims to evaluate:
► the prevalence of PEI in patients with pancreatic ductal adenocarcinoma and PanNETs (henceforth termed pancreatic malignancy);
► the most appropriate diagnostic strategy;
► the impact of adequate diagnosis and treatment of PEI on patient treatment and outcomes.

## STUDY DESIGN

The study will be conducted in two steps, as summarised in figure 1.

Step 1: a prospective cross-sectional assessment of the prevalence of PEI-related symptoms in patients with pancreatic malignancy (this will be termed 'the demographic cohort'). A separate cohort of patients will be tested to elucidate the most efficient diagnostic panel for PEI in pancreatic malignancy (this will be termed 'the diagnostic cohort').

Step 2: a prospective longitudinal validation of the diagnostic panel designed and tested in step 1 and evaluation

**Step 1**
Diagnostic panel design

**The demographic cohort**

N=50 evaluable patients

Assessed at baseline for symptoms of pancreatic enzyme insufficiency (PEI) and nutritional status using data from a standard of care panel of blood tests

**The diagnostic cohort**

N=25 patients

Assessed at baseline for symptoms of PEI and nutritional status using data from a standard of care panel of blood tests and a faecal elastase measurement. A breath test will test for PEI.

**Step 2**
Diagnostic panel evaluation

**The follow-up cohort**

N=50 evaluable patients

Pancreatic enzyme replacement therapy (PERT) prescribed (as req)

PEI diagnostic panel from Step 1 (diagnostic cohort) will be validated

**Figure 1** Study design overview.

| | Demographic cohort N = 50 | Diagnostic cohort N = 25 | Follow-up cohort N = 50 |
|---|---|---|---|
| Primary objectives | Prospective assessment of PEI prevalence | Design and evaluate an optimal PEI diagnostic panel | Prospective evaluation of the designed diagnostic panel |
| Secondary objectives | To determine the prevalence of PEI-related symptoms at first oncological referral. To assess the proportion of patients receiving PERT at the time of oncological referral. To evaluate nutritional status of patients at the time of oncological referral (using data from a panel of 'standard of care' blood tests, weight, Body Mass Index [BMI], Mid-Upper Arm Circumference [MUAC] (reflects both fat mass and fat-free mass), handgrip strength (Measurement of upper body function) and Stair Climb test [SC-test} (to calculate stair climb power)). To evaluate anorexia at baseline by using the FAACT-A/CS (with VAS) (functional Assessment of Anorexia Cachexia Tool (Anorexia Cachexia Scale) with Visual Analogue Scale). | In addition to secondary objectives for demographic cohort; To assess the feasibility of performing the PEI breath test and data from a standard of care faecal elastase-1 measurement. To assess, using the "acceptability questionnaire" (developed specifically for this study), the acceptability of these investigations by patients. | Percentage of patients who completed in full the proposed diagnostic panel. Percentage of patients with a positive experience of the diagnostic panel performed ("acceptability questionnaire"), developed specifically for this study. Median change in QoL, FAACT-A/CS (with VAS), BMI, weight, MUAC, handgrip strength and SC-test between baseline and follow-up assessment after 6 weeks, 3 months and 6 months of follow-up. Proportion of patients with PEI -related symptoms at baseline, 6 weeks, 3 months and 6 months from recruitment. Proportion of patients who have a normalisation of the diagnostic panel after 6 weeks, 3 months and 6 months from recruitment. Percentage of patients with good compliance (defined as taking at least 80% of the doses of PERT suggested by dietitian). Percentage of patients who develop any grade PERT-related toxicities. Percentage of patients with a positive experience of the dietetic intervention provided ("feedback questionnaire"), developed specifically for this study. Median OS of the whole population. Percentage of patients starting anticancer therapy. Median dose intensity of the received anti-cancer therapy. Correlation between radiological findings and nutritional status measurements. |

**Figure 2** Study objective. PEI, pancreatic exocrine insufficiency; PERT, pancreatic enzyme replacement therapy.

of dietitian intervention (including PERT) and its impact on weight loss, symptom evolution, chemotherapy dose-intensity, QoL and OS (this will be termed 'the follow-up cohort').

## STUDY OBJECTIVES AND PATIENT ELIGIBILITY

A summary of study objectives are provided in figure 2.

### Demographic cohort

The primary objective is to prospectively investigate the prevalence of PEI in patients with inoperable pancreatic malignancy. Prevalence will be determined by the presence of symptoms deemed in keeping with PEI by the research dietitian; alongside the absence of other causes for symptoms or standard diagnostic techniques (FE-1).

Secondary objectives include the following, at baseline oncological appointment:

▶ To assess the proportion of patients receiving PERT.
▶ To evaluate nutritional status (using a panel of blood tests (including nutritional parameters), weight, body mass index (BMI), mid-upper arm circumference (MUAC) (reflects both fat mass and fat-free mass), handgrip strength (measures upper body function) and stair climb test (SC-test) (to calculate stair climb power[36 37])).
▶ To evaluate anorexia, using the Functional Assessment of Anorexia/Cachexia Therapy questionnaire (FAACT-A/CS) and Visual Analogue Scale (VAS).[38 39]

Eligible patients for the demographic cohort are those who have biopsy-proven or clinically suspected (by specialist multidisciplinary team (MDT) meeting) inoperable (locally advanced or metastatic) pancreatic ductal adenocarcinoma (and variants) or PanNET. There is no minimal time-frame for patients to have been diagnosed with cancer. Patients must be ≥18 years and able to provide written, informed consent and are being considered for

first-line chemotherapy. Patients with PanNET may have received previous systemic treatment, but cannot be on active treatment.

Patients are deemed ineligible if they have had previous gastric, duodenal or pancreatic resections, if they have an intolerance/aversion to pork-containing products for religious or personal reasons. Additionally, patients are ineligible if they have comorbidities that increase the probability of PEI, including but not limited to chronic pancreatitis,[25] cystic fibrosis,[40] coeliac disease,[41] inflammatory bowel disease,[42 43] diarrhoea-dominant irritable bowel syndrome,[44] diabetes diagnosed >5 years ago.[45–47]

### Diagnostic cohort

The primary objective is to design and evaluate an optimal diagnostic panel to detect PEI in patients with inoperable pancreatic malignancy.

In addition to the secondary objectives of the demographic cohort:

▶ To assess the feasibility and acceptability (using a specifically designed 'Acceptability Questionnaire') of the [13]C-MTBT and the FE-1 test.

Eligible patients for the diagnostic cohort are those who fulfil the eligibility criteria for the demographic cohort. In addition, patients with potentially operable disease but who have not undergone surgery for whatever reason (i.e., comorbidities) would be eligible if all other eligibility criteria are met. Additionally, patients diagnosed with adenocarcinoma (and variants) will be allowed to have received previous systemic treatment but will be required to be off active treatment for a minimum of 3 months to be included in this cohort.

In addition to the exclusion criteria for the demographic cohort, patients must not be allergic to metoclopramide, a prokinetic used in the [13]C-MTBT.

### Follow-up cohort

The primary objective is to prospectively evaluate the proposed PEI diagnostic panel in a cohort of patients with inoperable pancreatic malignancy.

Secondary objectives include the following:

► To evaluate the feasibility of applying the designed diagnostic panel in clinical practice and to assess the patient acceptability of these investigations.
► To quantify changes to nutritional status (BMI, weight, MUAC, handgrip and SC-test), evaluate symptoms and extent of diagnostic panel normalisation, anorexia (FAACT-A/CS (with VAS)) and QoL at 6 weeks, 3 months and 6 months after recruitment and dietetic input.
► To evaluate PERT compliance and toxicity.
► To assess patient perceptions of the dietetic care provided.
► To evaluate the impact of dietetic intervention on OS, anticancer therapy starting rate and anticancer therapy dose intensity.
► Exploratory radiological surrogates (i.e., intra-abdominal fat or psoas muscle measurements) from standard of care CT scans and its correlation with nutritional assessment may be investigated.
► To evaluate the median dose intensity of the received anticancer therapy and the correlation between radiological findings and nutritional status measurements.[48]

Eligible patients for the follow-up cohort are those fulfilling the eligibility criteria for the diagnostic cohort and if further follow-up at The Christie NHS Foundation Trust is planned.

## CLINICAL ASSESSMENTS

Consecutive patients with inoperable pancreatic malignancy referred for consideration of first-line systemic therapy will be considered for eligibility. Eligible patients will be provided with verbal and written study information and given sufficient time to consider participation.

Clinical assessments will be undertaken as per figure 3. All consenting patients will undergo a prospective assessment of nutritional status, will be screened for PEI and will receive tailored advice by the research dietitian in established oncology clinics.

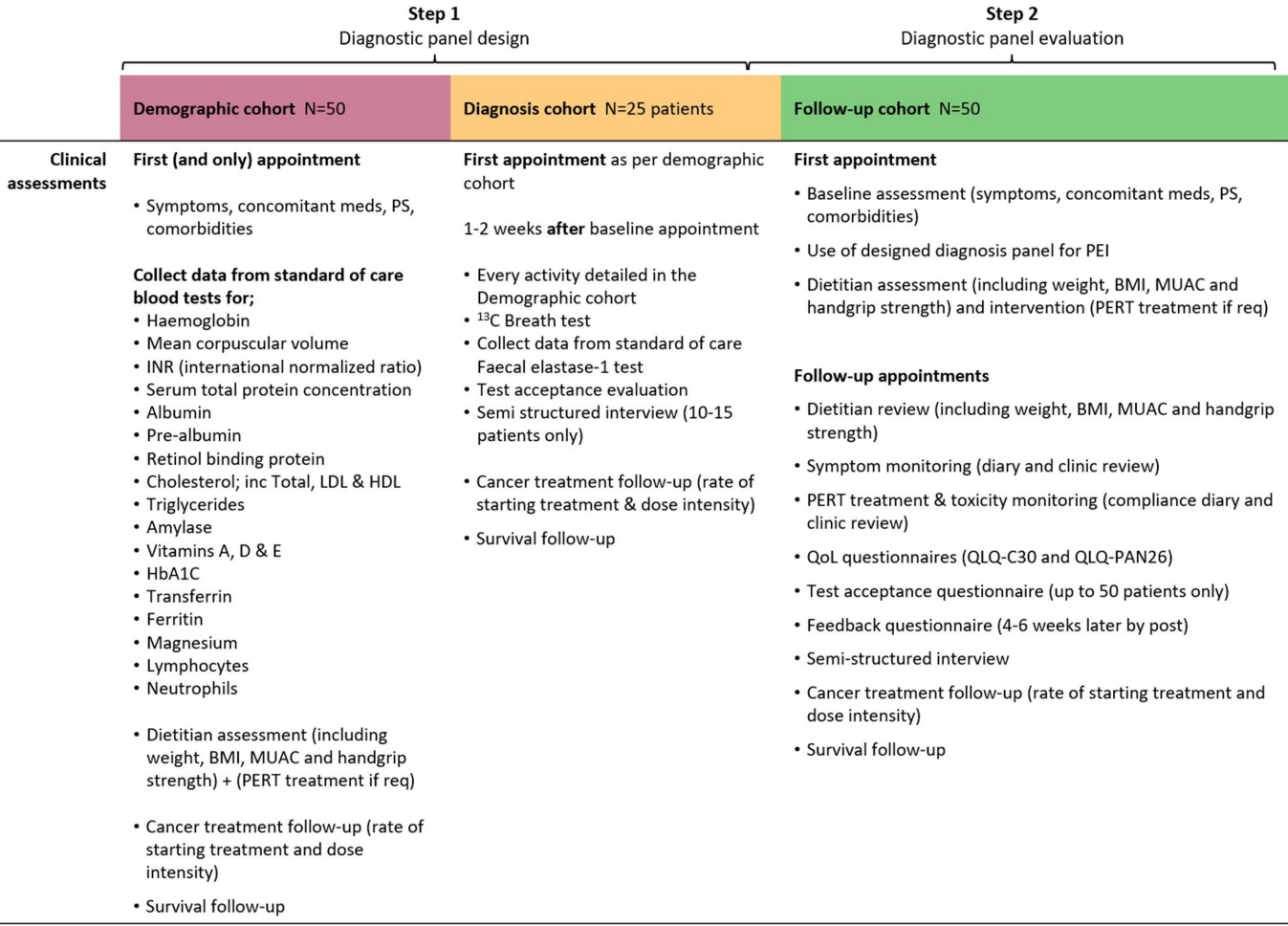

**Figure 3** Clinical assessment by cohort. Patients in the follow-up cohort will be reviewed (at week 6, month 3 and month 6 from study entry) by the study dietitian for further intervention and assessment. BMI, body mass index; HbA1C, haemoglobin A1C; HDL, high-density lipoprotein; LDL, low-density lipoprotein; MUAC, mid-upper arm circumference; PEI, pancreatic exocrine insufficiency; PERT, pancreatic enzyme replacement therapy; PS, performance status; QoL, quality of life.

## Demographic cohort

### Screening and visit 1

At baseline, patients will be screened against inclusion criteria. Written, informed consent must be granted before patients are registered. If appropriate, the baseline assessment can be performed on the same day as screening.

Assessments undertaken are as follows (figure 3):

► Physical examination; vital signs, height, weight, BMI, MUAC, handgrip strength, SC-test and FAACT-A/CS (with VAS).
► Baseline symptoms (PEI-related) and graded as per Common Terminology Criteria for Adverse Events (CTCAE) V.4.03.[49]
► PEI-relevant concomitant medications.
► ECOG-PS.
► Blood collection for nutritional panel.
► Dietitian assessment and counselling (PERT will be commenced, if required).
► PERT treatment and toxicity assessment (if patient on PERT).

### Follow-up visits

No follow-up visits will be required. Beyond study participation, dietetic input will be provided as per standard of care outside the context of this study. Information on subsequent chemotherapy treatment (starting rate and dose intensity) and survival outcomes will be collected.

## Diagnostic cohort

### Screening and visit 1

The baseline assessment will be completed as per the 'demographic cohort'.

### Visit 2 (prior to starting systemic treatments)

The following assessments will be performed:

► Weight.
► FE-1 test (container provided at baseline visit and returned at visit 2).
► $^{13}$C-MTBT (takes around 6 hours to complete, including administering bread spread with $^{13}$C butter and subsequent collection of the patient's breath in small breath bags at timed intervals, which will be analysed for $^{13}$C quantity).
► Acceptability Questionnaire for FE-1 test and $^{13}$C-MTBT (all patients) to provide opinions on the burden that these extra tests may add.

### Follow-up visits

No follow-up visits will be required. Patients attending clinic for further follow-up/treatment will have further dietetic input, as required, outside of the context of this study. Information on the subsequent chemotherapy (starting rate and dose intensity) and survival outcomes will be collected.

## Follow-up cohort

Prior to opening recruitment to this cohort, data from the demographic and diagnostic cohorts will be analysed, which will dictate the most informative diagnostic panel devised to be used in the follow-up cohort.

### Screening and visit 1

Screening will be completed as per the demographic and diagnostic cohorts.

In addition, further assessments (figure 3) include:

► QoL questionnaires (QLQ-C30 (all patients) and either QLQ-PAN26 (pancreatic ductal adenocarcinoma) or QLQ NET-21 (PanNET)).
► A symptom and PERT diary for data collection will be provided.

### Visit 2 (within 2 weeks)

► Designed PEI diagnostic panel (from all the potential combinations of FE-1 test, symptom assessment, nutritional assessment (weight, BMI, MUAC, handgrip strength, SC-test, FAACT-A/CS (with VAS) and nutritional blood panel)), as per findings from the diagnostic cohort).
► 'Acceptability Questionnaire' regarding the burden that this diagnostic panel added.

### Week 4–6 from study entry

► 'Feedback questionnaire' regarding perception of dietetic input (all patients) (posted to the patient and returned using a provided stamped-addressed envelope).

### Follow-up visits

At 6 weeks, 3 months and 6 months after recruitment:

► Weight, BMI, MUAC, handgrip strength, SC-test and FAACT-A/CS (with VAS).
► Physical examination (symptom directed), including vital signs (if appropriate).
► ECOG-PS.
► Dietitian assessment and counselling, nutrition support advice (diet, nutritional supplements, etc.) and PERT, as required.
► PERT treatment review and toxicity assessment (if taking PERT).
► Symptom and PERT diary collection.
► QoL questionnaires repeated as per visit 1.
► Survival and chemotherapy treatment monitoring (retrospectively).

Patients attending clinic for further follow-up/treatment will be provided with dietetic input, as required, outside of the context of this study.

## STATISTICAL ANALYSIS

### Sample size

No formal sample size calculation was performed. Instead, a realistic estimation of the number of patients possible to recruit was made using established referral rates and the length of time this study will recruit for. A high drop-out rate was expected due to the poor outcomes of patients with pancreatic cancer. Therefore, sufficient patients will be recruited to ensure the planned number of 'evaluable

patients' for each cohort is reached. These are defined as follows:

### Demographic cohort
Up to 50 eligible patients completing the assessment required in 'visit 1'.

### Diagnostic cohort
Up to 25 eligible patients willing (at time of consent) to complete the breath test and other cohort-dependent examinations.

### Follow-up cohort
Up to 50 eligible patients completing assessments up to and including 'follow-up visit week 6'.

### Handgrip strength measurements contribute to the statistical calculation
In order to gather supporting data, a non-selected sample of 12 patients with pancreatic ductal adenocarcinoma performed the handgrip test as per standard of care assessment at The Christie (mean percentile was 75; SD 16). Using these data, the follow-up cohort (50 patients with handgrip assessment at baseline and at 6 weeks) will have a power of 0.75 to show an improvement from 75 to 82 (7-point improvement). This is assuming an alpha error of 0.15 and same SD in both baseline and 6-week assessment handgrip results (SD 16). This is supporting evidence that the sample size for this study will be able to provide meaningful and robust results.

### Study end points
The primary end points of the three cohorts are as follows:

#### Demographic cohort
► Proportion of patients with symptoms/findings in keeping with a PEI diagnosis.

#### Diagnostic cohort
► Odds ratio for prediction of diagnosis of PEI (measured by $^{13}$C-MTBT) of the most accurate diagnostic panel (designed from all potential combinations of FE-1 test, symptom assessment, nutritional assessment (weight, BMI, MUAC, handgrip strength, SC-test, FAACT-A/CS (with VAS) and nutritional blood panel).

#### Follow-up cohort
► Rate of PEI diagnosis according to the designed diagnostic panel (diagnostic cohort).

### Data analysis
Frequency tables for all categorical variables, arranged by category, will be produced for comparison. Continuous variables (age, weight, BMI, MUAC, handgrip strength, SC-test and FAACT-A/CS (with VAS) will be presented, using the median and range (minimum, maximum) or mean (variance), depending on whether data distribution appears symmetrical.

For exploratory analyses, means will be compared using either Student's t-test (if parametric validity conditions are fulfilled) or non-parametric Wilcoxon-Mann-Whitney U tests. Proportions will be compared using either $\chi^2$ statistics or Fisher's exact test, as appropriate. Toxicity data will be tabulated. The worst toxicity grade over all cycles according to the CTCAE V.4.03[49] will be reported.

Median survival will be calculated using the Kaplan-Meier estimator technique. Median OS will be displayed with the 95% CI. For comparison of survival curves, log-rank test will be applied. Multivariable analyses will also be performed (Cox regression).

Analysis of data collected from the demographic and diagnostic cohorts will be undertaken to devise the optimal diagnostic panel, using $^{13}$C-MTBT as a reference to diagnose PEI. Results from the breath test will be reported as a dichotomised variable (normal or abnormal).

Logistic regression will be performed, aiming to choose the most informative, but simplest panel of tests, to predict PEI as the $^{13}$C-MTBT has done.

Individually measured blood parameters, together with other calculated scores (such as, but not limited to the 'prognostic nutritional index' (combining lymphocytes and albumin)) will be included in such analysis, if required.

Further analysis on the completion of the follow-up cohort will evaluate the panel's accuracy and acceptability for use in clinical practice.

### ETHICS AND DISSEMINATION
Full ethical approval has been granted by the North West Greater Manchester East Research and Ethics Committee, reference: 17/NW/0597, favourable opinion granted 7 December 2017. This manuscript reflects the latest protocol V.8 approved 21 April 2020.

The study will be conducted according to the principles of Good Clinical Practice, General Data Protection Regulation and Data Protection Act 2018 for Health and Care Research. The sponsor and study team will ensure approval of the study protocol, participant information sheets, consent forms, letters to general practitioners and supporting documents by the appropriate regulatory body and research and ethics committee prior to participant recruitment. Documents will be stored securely with restricted access for at least 15 years.

Written, informed consent will be obtained from each patient, and an identification number provided. Any published data will not contain personally identifiable data. Findings will be disseminated by presentation at national/international conferences, publication in peer-review journals and distribution via patient advocate groups.

### PATIENT AND PUBLIC INVOLVEMENT
Patient advocate groups (Pancreatic Cancer UK and Neuroendocrine Cancer UK (formerly known as the

NET Patient Foundation)) were involved in the development of this study protocol. Results will be disseminated to patients via these advocate groups once results are available.

**Author affiliations**
¹Nutrition & Dietetics, The Christie NHS Foundation Trust, Manchester, UK
²Medical Oncology Department, The Christie NHS Foundation Trust, Manchester, UK
³Division of Cancer Sciences, The University of Manchester, Manchester, UK
⁴Pancreatic Cancer UK, London, UK

**Acknowledgements** The authors would like to thank the patient advocate groups (Pancreatic Cancer UK and Neuroendocrine Cancer UK (formerly known as the NET Patient Foundation)) for helping in the development of this study protocol.

**Contributors** The protocol was devised by AL, LM, AB and MA. JWV supervised the development of the study protocol and approved the final version. JB provided independent statistical support. KV coordinated study set-up and supervised the study as project manager. LC is responsible for patient recruitment and dietetic assessment. JWV, RH, MGM, ZAK and AL, are responsible for identifying and discussing with potentially eligible patients. All authors approved the manuscript. Peer review was undertaken during protocol development, and the study has been adopted by the National Cancer Research Institute Upper GI Clinical Studies Group (Pancreatic subgroup).

**Funding** AL has received funding from European Society for Medical Oncology (ESMO) Fellowship Programme, Pancreatic Cancer Research Fund, Spanish Society of Medical Oncology (SEOM) Fellowship Programme, American Society of Clinical Oncology (ASCO) Conquer Cancer Foundation Young Investigator Award and The Christie Charity. ZAK was funded by The Christie Charity. LC was funded by Pancreatic Cancer UK Fellowship (Clinical Pioneer Award 2015) and Neuroendocrine Cancer UK, formerly known as the NET Patient Foundation. KV is funded by Cancer Research UK (Funder reference: C2930/A25234, University of Manchester reference: R120976).

**Competing interests** LC reports grants from Pancreatic Cancer UK andNeuroendocrine Cancer UK (formerly known as the NET Patient Foundation), during the conduct of the study; travel and educational support from Mylan and Ipsen, outside the submitted work; AL reports grants and personal fees: travel and educational support from Ipsen, Pfizer, Bayer, AAA, SirtEx, Novartis, Mylan and Delcath. Speaker honoraria from Merck, Pfizer, Ipsen, Incyte and AAA. Advisory honoraria from EISAI, Nutricia Ipsen, QED and Roche. Member of the Knowledge Network and NETConnect Initiatives funded by Ipsen. All outside the submitted work; ZAK reports educational and travel support from EISAI, outside the submitted work; MGM reports grants from NuCana, grants from Servier, grants from Ipsen, other from Novartis, outside the submitted work; RH reports personal fees from Ipsen, Mylan, Celgene and PrimeOncology, outside the submitted work; MA reports travel and educational support from Mylan, outside the submitted work; JWV reports personal fees from AstraZeneca, Debiopharm, Delcath Systems, Genoscience Pharma, Imaging Equipment Limited, Incyte, Ipsen, Keocyt, Merck, Mundipharma EDO, Novartis, PCI Biotech, Pieris Pharmaceuticals, QED, Wren Laboratories and Agios; grants, personal fees and non-financial support from NuCana, personal fees and non-financial support from Pfizer, grants and personal fees from Servier, outside the submitted work; KV, LM, AB and JB have nothing to declare.

**Patient and public involvement** Patients and/or the public were involved in the design, or conduct, or reporting, or dissemination plans of this research. Refer to the 'Patient and Public Involvement' section for further details.

**Patient consent for publication** Not required.

**Provenance and peer review** Not commissioned; externally peer reviewed.

**ORCID iD**
Lindsay E Carnie http://orcid.org/0000-0002-4626-3004

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
