## [Reviewer comments · BMJ Open]

ARTICLE DETAILS

TITLE (PROVISIONAL)	Prospective observational study of prevalence, assessment and treatment of pancreatic exocrine insufficiency in patients with inoperable pancreatic malignancy (PANcreatic cancer Dietary Assessment - PanDA): a study protocol
AUTHORS	Carnie, Lindsay; Lamarca, Angela; Vaughan, Kate; Kapacee, Zainul Abedin; Mccallum, Lynne; Backen, Alison; Barriuso, Jorge; McNamara, Mairéad; Hubner, R; Abraham†, Marc; Valle, Juan

VERSION 1 – REVIEW

REVIEWER	Chris Forsmark University of Florida, USA
REVIEW RETURNED	09-Aug-2020

GENERAL COMMENTS	This planned observational study will utilize 3 separate cohorts. One cross-sectional cohort will assess for the prevalence of clinical or laboratory features related to pancreatic exocrine insufficiency in 50 patients with unresectable pancreatic adenocarcinoma or neuroendocrine tumors. A second cohort of 25 subjects, with inoperable cancer or operable cancer awaiting surgery, will be used to test for the presence of pancreatic exocrine insufficiency by performing a mixed triglyceride breath test. Finally, a prospective cohort of 50 subjects with inoperable pancreatic cancer will be treated with pancreatic enzyme replacement therapy, with sequential assessment of tolerability, quality of life, and nutritional indices. This study addresses an important clinical problem, and has the potential to inform and improve clinical practice. A number of potential pitfalls may be worth considering as the trial design is finalized. 1. It seems, at least to this reviewer, that the inclusion of patients with neuroendocrine tumors will introduce significant bias. Unlike those with adenocarcinoma, these patients rarely develop exocrine insufficiency, and have prolonged survival. They also often are treated with octreotide, which will change pancreatic enzyme secretion. I do not understand the rationale for including this subgroup. Was it the intention that they would be analyzed separately? Eliminating this subgroup will not harm patient accrual much as they are so rare, and would make the groups more harmonized and comparable.2. The sample size is arbitrary, and may not be sufficient to achieve the stated Aims. In terms of handgrip strength, your sample size assumes enzyme therapy may improve strength, but the main effect is likely to be reducing loss of strength, rather than improving strength.3. Will there be a protocol for providing dietary guidance to these subjects in the follow-up cohort? Will there be a protocol for dosage and instructions on enzyme therapy? To eliminate some
---

	variability, the protocol should provide standardized approaches to these two issues. 4. Will you be measuring muscle mass on CT scans? You note an exploratory analysis of intra-abdominal fat, but did not mention psoas muscle measurements. 5. Will the follow-up cohort undergo breath testing? If not, why not?
--	--

REVIEWER	Keith Roberts University Hospitals Birmingham, UK
REVIEW RETURNED	28-Sep-2020

GENERAL COMMENTS	Many thanks for asking my thoughts. This is an important topic and the research is welcomed. The UK national prospective audit of pancreatic cancer, RICOCHET, will show wide variation in the use of PERT, being lowest among those with unresectable cancer. Pancreatic malignancies is frequently written in the text but I think pancreatic malignancy is the correct grammar. There are three components to the study, conducted over two steps. Step 1 will determine the prevalence of PEI related symptoms – a demographic cohort of 50 patients. A second component will determine the optimal diagnostic test among a diagnostic cohort of 25 patients. Step 2 comprises a validation study of the diagnostic tests from step 1 with evaluation of dietitian intervention and its impact upon weight loss, symptom evolution and other relevant end points. It is unclear if there is, and if so, what the comparator group will be for the patients in step 2. The change in QoL and other repeated measures will be undertaken compared to baseline. If there is no comparator group then changes over time will not only relate to treated PEI but also progression of disease and treatment. If they are patients already treated within the Christie where patients are likely to have excellent care which will likely include PERT and dietician involvement then there may be limited scope for an optimised pathway to have impact. A better comparator group would be a cohort of advanced cancer patients treated at another centre. It would clearly not be possible for the Christie team to change their practice to avoid PERT, particularly given the NICE guidance in this area. This is always a challenge for studies of PERT in pancreatic cancer. Though many patients do not receive PERT, it is typically considered unethical to withhold PERT, despite a lack of high level evidence. It may be that patients without PEI or with mild PEI can be used as a comparator. It is unclear why the PEI-Q tool is not being used. It may be because the design of the study predates publication of the PEI-Q. If so the researchers may wish to add that tool to the study The two components of Step 1 comprise research elements that will yield data that is unlikely to be novel as these topics have been studied previously. However, there is quite wide variation in the reported prevalence of PEI among these cohorts with previous studies often including a more diverse cohort. It is clearly essential to have a baseline understanding of prevalence, and to gain experience of the c13 breath test and so this Step is important.
---

	The study will yield clinically relevant data and, importantly, will highlight the need to consider PEI and treatment with PERT.
--	--

VERSION 1 – AUTHOR RESPONSE

Reviewer(s)' Comments to Author:

Reviewer: 1

This study addresses an important clinical problem, and has the potential to inform and improve clinical practice. Reply: we are grateful for this feedback

A number of potential pitfalls may be worth considering as the trial design is finalized.

1. It seems, at least to this reviewer, that the inclusion of patients with neuroendocrine tumors will introduce significant bias. Unlike those with adenocarcinoma, these patients rarely develop exocrine insufficiency, and have prolonged survival. They also often are treated with octreotide, which will change pancreatic enzyme secretion. I do not understand the rationale for including this subgroup. Was it the intention that they would be analyzed separately? Eliminating this subgroup will not harm patient accrual much as they are so rare, and would make the groups more harmonized and comparable. Reply: we fully agree that NETs and PDAC are different, with different PEI-derived problems; rationale to include NET patients relies on source of funding supporting this study. At time of study design, we felt that some of the main findings would be transferable to the NET patients as well. In view of the fact that NET are rare, we expect majority of study population to be in the form of PDAC, thus we expect the inclusion of NET to have a minimal impact on main findings

2. The sample size is arbitrary, and may not be sufficient to achieve the stated Aims. Reply: this has been added to limitations sections In terms of handgrip strength, your sample size assumes enzyme therapy may improve strength, but the main effect is likely to be reducing loss of strength, rather than improving strength. Reply: this is correct and the most likely scenario, however, we would like to (ideally) see an improvement following intervention of PEI and dietitian input

3. Will there be a protocol for providing dietary guidance to these subjects in the follow-up cohort? Will there be a protocol for dosage and instructions on enzyme therapy? To eliminate some variability, the protocol should provide standardized approaches to these two issues. Reply: Only two dietitians have been / currently are involved in the care of the patients recruited in this study, thus advice provided has been standardized. In addition, a specific protocol for PEI-management is available in our institution; it is currently under consideration as a separate publication.

4. Will you be measuring muscle mass on CT scans? You note an exploratory analysis of intra-abdominal fat, but did not mention psoas muscle measurements. Reply: exploring sarcopenia is one of the future post-hoc research questions, specifics on methodology have not been defined as yet; this has been added to the current wording in the protocol

5. Will the follow-up cohort undergo breath testing? If not, why not? Reply: breath test was only planned to be used as gold-standard for diagnosis of PEI in the diagnostic cohort. In view of burden that performing breath test represents for patients (fasting of around 12 hours in total, 6 hours of test in hospital and diet adjustment), it was felt too much of a burden for the rest of cohorts or for considering this “standard-of-care”.

Reviewer: 2

Many thanks for asking my thoughts. This is an important topic and the research is welcomed. The UK national prospective audit of pancreatic cancer, RICOCHET, will show wide variation in the use of PERT, being lowest among those with unresectable cancer. Reply: we are grateful for this feedback

Pancreatic malignancies is frequently written in the text but I think pancreatic malignancy is the correct grammar. Reply: this has been corrected through the text

It is unclear if there is, and if so, what the comparator group will be for the patients in step 2. The change in QoL and other repeated measures will be undertaken compared to baseline. If there is no comparator group then changes over time will not only relate to treated PEI but also progression of

disease and treatment. If they are patients already treated within the Christie where patients are likely to have excellent care which will likely include PERT and dietician involvement then there may be limited scope for an optimised pathway to have impact. A better comparator group would be a cohort of advanced cancer patients treated at another centre. It would clearly not be possible for the Christie team to change their practice to avoid PERT, particularly given the NICE guidance in this area. This is always a challenge for studies of PERT in pancreatic cancer. Though many patients do not receive PERT, it is typically considered unethical to withhold PERT, despite a lack of high level evidence. It may be that patients without PEI or with mild PEI can be used as a comparator. Reply: there is no pre-planned cohort to be used for direct comparison; however, we do have previous publication in this field with historical cohort which will help to put our findings into perspective [McCallum L, Lamarca A, Valle JW. Prevalence of symptomatic pancreatic exocrine insufficiency in patients with pancreatic malignancy: nutritional intervention may improve survival. Cancer Research Frontiers. 2016;2(3):352-367]. Comparison will be performed inter-patient compared to baseline, and findings can be adjusted to response to therapy to adjust for confusion introduced by that fact that “changes over time will not only relate to treated PEI but also progression of disease and treatment”

It is unclear why the PEI-Q tool is not being used. It may be because the design of the study predates publication of the PEI-Q. If so the researchers may wish to add that tool to the study Reply: we thank the reviewer for this comment; this tool was not available when the study was first designed in 2016-2017. The study has almost completed recruitment now so it would not be possible for this to be incorporated now, unfortunately.

The two components of Step 1 comprise research elements that will yield data that is unlikely to be novel as these topics have been studied previously. However, there is quite wide variation in the reported prevalence of PEI among these cohorts with previous studies often including a more diverse cohort. It is clearly essential to have a baseline understanding of prevalence, and to gain experience of the c13 breath test and so this Step is important. The study will yield clinically relevant data and, importantly, will highlight the need to consider PEI and treatment with PERT. Reply: we are grateful for this feedback

VERSION 2 – REVIEW

REVIEWER	Keith Roberts University Hospitals Birmingham
REVIEW RETURNED	28-Nov-2020
GENERAL COMMENTS	Thank you for addressing my comments. I have no more. This is an important body of work in a field that is lacking much important detail.